# BT-MoE: A Budget-Aware Tuning Framework for Joint Bit–Rank Allocation in MoE Models

## Abstract

Quantization is a critical approach for efficiently deploying Mixture-of-Experts (MoE) models with massive parameters. However, MoE models suffer from non-negligible accuracy loss with extreme quantization, such as under 4 bits. To address this, we introduce BT-MoE, a novel framework that achieves a unified and globally optimal allocation of mixed-precision bit-widths and low-rank compensator configurations. Our key insight is to formalize this co-design problem as a Multiple-Choice Knapsack Problem (MCKP). To make this NP-hard problem computationally feasible, we further propose an efficient proxy metric based on layer-wise quantization loss for rapid configuration impact assessment, so that a standard Integer Linear Programming (ILP) solver can solve the MCKP within a practical time. Our comprehensive evaluation demonstrates that BT-MoE consistently outperforms state-of-the-art quantization methods across various MoE models and benchmarks. By systematically exploring the design space, BT-MoE achieves superior accuracy-memory trade-offs, significantly improving the deployability of large MoE models on resource-constrained hardware.

## 1 Introduction

The Mixture-of-Experts (MoE) architecture has emerged as the dominant paradigm for scaling Large Language Models (LLMs), achieving state-of-the-art performance by replacing dense MLP blocks with specialized expert networks and dynamic routing mechanisms (Jiang et al., 2024; Qwen et al., 2025; Liu et al., 2024; Fedus et al., 2022). This design delivers enhanced model capacity without sacrificing computational efficiency. However, these advantages come at the cost of critical deployment challenges. The memory footprint of MoE models is typically several times larger than dense counterparts. For instance, Mixtral-8x7B requires approximately 88 GB of memory, which exceeds the 24 GB capacity of consumer GPUs like RTX 4090, while DeepSeek-V3's 671B parameters surpass the memory capacity of eight H100 GPUs (Liu et al., 2024).

Deploying MoE models that exceed single-GPU memory requires either offloading to host or external memory, which introduces additional communication overhead, or resorting to expensive multi-GPU inference (Eliseev & Mazur, 2023; Rajbhandari et al., 2022). Among various compression techniques, model quantization has emerged as the most promising approach for LLM deployment (Frantar et al., 2022; Xiao et al., 2023; Badri & Shaji, 2023). Mixed-precision quantization further enhances MoE model performance by assigning different bit-widths to model components based on their quantization sensitivity (Huang et al., 2025b; Tang et al., 2024). This approach allocates lower bit-widths to quantization-robust components while preserving higher precision for sensitive ones.

Despite these advances, a critical limitation emerges under aggressive compression scenarios. As shown in Figure 1, while existing methods like GPTQ (Frantar et al., 2022) and HQQ (Badri & Shaji, 2023) maintain reasonable accuracy at 4-bit precision, they suffer severe performance degradation when pushed to 3-bit quantization. For instance, Mixtral-8×7B experiences a perplexity increase from 3.70 (FP16) to 4.73 (3-bit GPTQ), indicating a severe degradation in model performance.

To address this accuracy degradation, recent works have introduced low-rank compensators as post-quantization correction mechanisms (Li et al., 2025a; Huang et al., 2025a). These methods capture quantization residuals using low-rank matrix factorization, effectively recovering lost information. However, existing approaches suffer from a fundamental design flaw: they treat bit-width selection

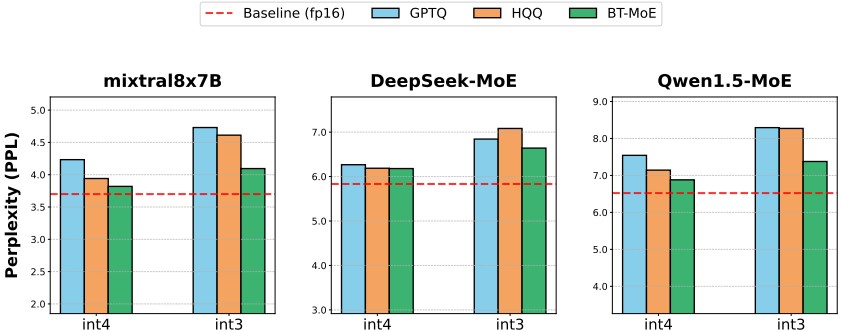

Figure 1: Comparison of existing quantization methods and BT-MoE on various bit-precision.

and compensator rank allocation as independent optimization problems, typically applying uniform configurations across all experts.

This uniform approach fundamentally misaligns with the heterogeneous nature of MoE architectures. Our empirical analysis reveals that different experts exhibit dramatically different sensitivities to quantization: some experts can tolerate aggressive 2-bit quantization with minimal accuracy loss, while others require higher precision even with compensator assistance. More critically, we discover a complex interdependency between bit-width and compensator rank: the optimal compensator rank for an expert is not fixed but depends heavily on its quantization bit-width, and vice versa. This interdependency creates a vast combinatorial optimization space that defies simple heuristic solutions. For a typical MoE model with 64 experts, each having 5 bit-width options and 5 rank choices, the total configuration space exceeds $25^{64} \approx 10^{90}$ possibilities, making exhaustive search computationally intractable. Existing greedy approaches, such as allocating the highest resources to the most frequently activated experts, fail to capture the nuanced trade-offs between different configuration combinations and often converge to local minima.

To address these challenges, we propose BT-MoE, a novel framework that introduces a Budget-Aware Tuning approach for the joint allocation of mixed-precision bit-widths and low-rank compensators. Our central idea is to cast the joint design as a Multiple-Choice Knapsack Problem (MCKP), where each expert selects exactly one configuration from a set of (bit-width, rank) candidates under a global resource budget. Although the resulting problem is NP-hard, we make it computationally tractable by employing an efficient layer-wise proxy for quantization-induced degradation. This proxy enables rapid evaluation of thousands of candidate configurations without exhaustive full-model retraining or validation, allowing the global selection to be solved by a standard Integer Linear Programming (ILP) solver within a practical time.

Our contributions are threefold. (1) We identify and formalize the complex coupling between weight bit-width and compensator rank in MoE quantization. (2) We propose an ILP-based global allocation method that jointly optimizes both dimensions, enabled by an efficient proxy metric that makes the global optimization computationally feasible. (3) We demonstrate consistent and significant improvements over existing methods across multiple MoE models and benchmarks, achieving superior accuracy-memory trade-offs that enable practical deployment of large MoE models on resource-constrained hardware.

## 2 RELATED WORKS

**Post-Training Quantization (PTQ).** PTQ has become the standard for LLM compression, avoiding the prohibitive cost of Quantization-Aware Training (QAT). In this line of work, state-of-the-art methods such as GPTQ (Frantar et al., 2022) and AWQ (Lin et al., 2024) have successfully compressed dense LLMs to 4-bit precision without significant accuracy loss. More recently, calibration-free methods like HQQ (Badri & Shaji, 2023) have also been explored, which capture outliers using a Super-Laplacian distribution with a closed-form solution. However, a uniform quantization scheme is often suboptimal for models with heterogeneous components like MoEs. To address this, Mixed-precision quantization allocates varied bit-widths to model components based on their differing sensitivities to quantization error (Wang et al., 2019; Dong et al., 2019). This principle is

particularly effective for MoE models, whose experts naturally exhibit heterogeneous sensitivities (Huang et al., 2025b). Nevertheless, a key challenge persists: existing PTQ approaches, whether uniform or mixed-precision, still struggle to maintain accuracy when compressing MoE models to higher compression ratios, particularly in the sub-4-bit regime.

**Low-rank Compensation methods for LLM compression.** Low-Rank Factorization techniques are widely used to compensate for accuracy loss in LLM compression. ASVD (Yuan et al., 2023) proposes an activation-aware factorization method that uses a transformation matrix to absorb information about outliers from the activations into the weight matrices to compress the model. LoRC (Yao et al., 2024) applies low-rank factorization to the quantization error matrix and uses the low-rank matrices as a "compensator," making it an effective method for model accuracy recovery. SVDQuant (Li et al., 2025a) utilizes a high-precision, low-rank branch to absorb the most hard-to-quantize outliers in the weights and activations, thereby allowing the remaining, smoother residual component to be easily quantized to 4-bit precision. MiLo (Huang et al., 2025a) designs a mixed-rank compensation scheme specifically for MoE models, adaptively assigning varying ranks to different experts. Different from those efforts, we explore a joint optimization of mixed-precision quantization and low-rank compensation for different components in MoE models, systematically exploring the optimal model compression configuration.

## 3 METHODOLOGY

### 3.1 CO-DESIGN CHALLENGE IN MOE QUANTIZATION

The core challenge in compressing MoE models lies in their heterogeneous nature. Different components exhibit varying sensitivities to quantization, creating a complex optimization space. Our analysis, along with recent studies, reveals a clear hierarchy of these sensitivities (Duanmu et al., 2025). As illustrated in Figure 2a, we observe that attention layers are more sensitive than expert FFNs, and shared experts are more sensitive than regular experts. Furthermore, within a single MoE layer, experts exhibit varying sensitivities to quantization (Figure 2b). This heterogeneity motivates the use of mixed-precision quantization.

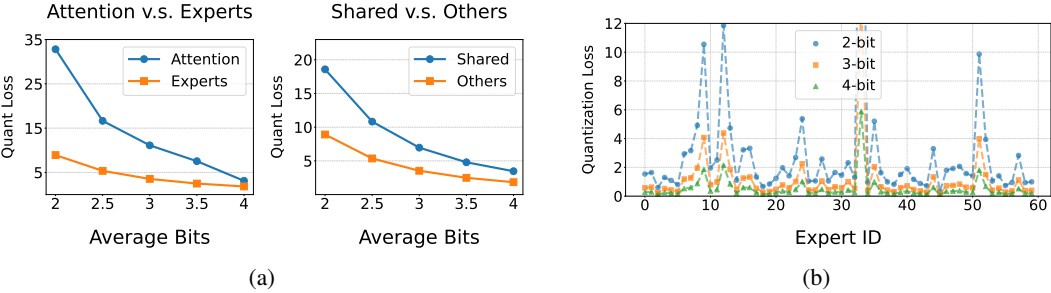

(a)            (b)

Figure 2: (a) Comparison of quantizing more bits for attention vs. experts and shared-experts v.s. others evaluated on the DeepSeek-MoE-16B-base model. (b) Quantization loss across experts in Qwen1.5-MoE's 1st layer under different bit width, group_size=128.

However, heuristic-based bit allocation (Li et al., 2025b) is insufficient, particularly under the aggressive compression required by strict memory budgets, which inevitably leads to significant quantization error. To counteract this residual error, low-rank compensators have been introduced as a powerful correction tool. It creates a second, coupled dimension for optimization: the rank of the compensator. However, the co-design of mixed-precision bit-widths and compensator ranks introduces three fundamental challenges:

**(1) Coupled Relationship: Nonlinear trade-offs between bit-width and rank.** The memory budgets for bit-width and rank compensation are interdependent. An expert quantized to a lower bit-width can have its accuracy recovered by a higher-rank compensator. We find that the benefit of increasing the compensator rank for an expert depends on its current quantization bit-width. For an expert quantized to 2-bit, a rank-32 compensator might provide a significant improvement; however, for an expert already at 4-bits, the same compensator may have little effect. As shown in Figure 3(a), a 4-bit configuration with no compensator might yield a similar quantization loss to a 3-bit

configuration with a rank-64 compensator, but the 4-bit configuration is more memory-intensive. This non-linear coupling between bit-width and rank indicates that optimizing one dimension in isolation will lead to a suboptimal outcome.

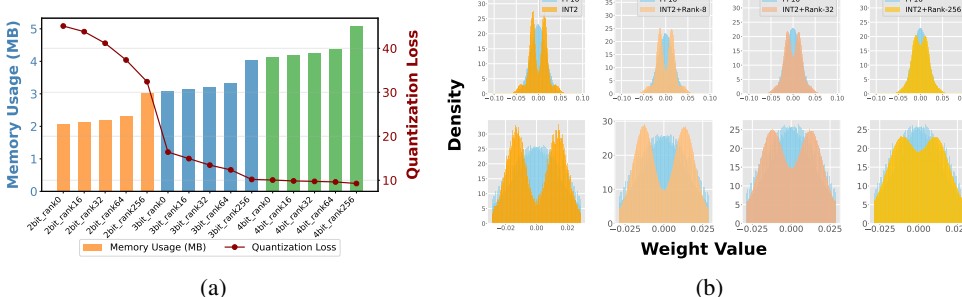

(a)          (b)

Figure 3: (a) Quantization Loss vs. Memory Footprint for a single expert (Layer 10, Expert 24) in DeepSeek-V2-Lite under various compression configurations. (b) Weight distribution for a single expert (Layer 20, Expert 17) in Qwen1.5-MoE under 2-bit quantization, illustrating the accuracy recovery from different compensator ranks.

**(2) Heuristic Limitations: Greedy and frequency-based allocation fall short.** Due to the coupled relationship described above, employing simple greedy algorithms or heuristic strategies can easily lead to a local optimum. For example, allocating the highest-cost configuration (i.e., the highest bit-width and rank) to the most sensitive expert might seem reasonable, but this can excessively consume the budget. This would leave dozens of other experts with poor configurations, consequently harming the overall model performance.

Some existing work on mixed-precision MoE quantization uses a heuristic based on expert activation frequency; however, we find this approach also fails to achieve optimal results. This is particularly true for MoE models like Mixtral-8x7B. This is because the model has a small number of experts per layer, leading to little variance in their activation frequencies, as shown in Figure 4a. For models with a large number of fine-grained experts, such as DeepSeek-MoE and Qwen1.5-MoE, the expert activation frequencies exhibit significant variance, as shown in Figure 4b and 4c. Consequently, for these models, a frequency-based mixed-precision allocation can be a reasonably effective strategy.

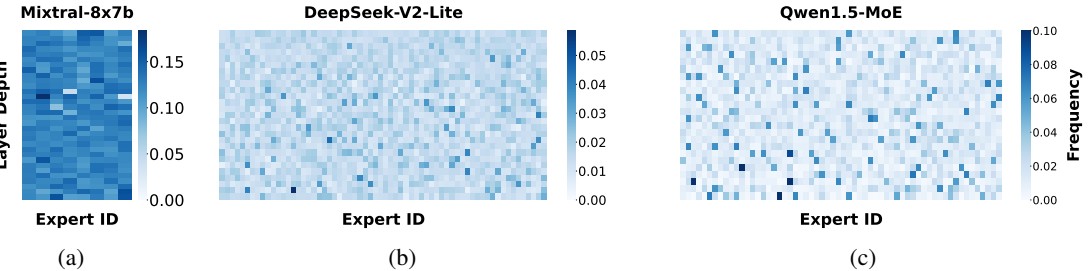

(a)          (b)          (c)

Figure 4: Heatmap of expert activation frequency in Mixtral8×7B, DeepSeek-V2-Lite and Qwen1.5-MoE on the WikiText-2 task. The vertical axis from top to bottom represents the layer depth, and the horizontal axis represents expert indices.

**(3) Combinatorial Explosion: Vast search space and costly evaluation.** A key difficulty in co-optimizing bit-width selection and compensator rank lies in the Combinatorial Explosion, which creates an extremely large design space. An MoE model contains a large number of experts, and each expert has multiple possible combinations of bit-widths and compensator ranks. For instance, models like Qwen1.5-MoE or DeepSeek-MoE have over 60 experts per layer. If we provide just a few bit-width options and several rank choices for each expert, every expert will have dozens of possible configurations. For the entire model, the total number of combined configurations is vast:

$$O\left(\left(|\mathcal{B}| \times |\mathcal{R}|\right)^{|\mathcal{E}|}\right),$$

where $\mathcal{B}$ is the number of bit-width options, $\mathcal{R}$ is the number of rank options, and $\mathcal{E}$ is the total number of experts in the model to be configured. making a brute-force search of all possibilities

computationally impossible. Therefore, this is a massive discrete combinatorial optimization problem that requires a systematic method to efficiently explore this vast design space.

This challenge is twofold. Beyond the sheer number of configurations, evaluating the quality of any single complete assignment by running a full model benchmark is also prohibitively expensive. To solve this, our framework introduces a two-part approach. First, to address the evaluation cost, we propose an efficient proxy metric based on layer-wise quantization loss. Through isolated perturbation experiments, we rapidly collect the quant-loss for each expert under every potential (bit, rank) configuration. This allows us to model the global accuracy degradation as the weighted sum of these local losses. Second, with the cost and impact of each configuration now quantified, we tackle the search problem by formulating this task as an Integer Linear Programming (ILP) problem. By minimizing the total weighted loss, the ILP solver can efficiently navigate the vast design space to find a provably optimal solution, bypassing the need for an exhaustive search. As our experiments demonstrate, this systematic approach yields configurations that are significantly superior to baseline methods.

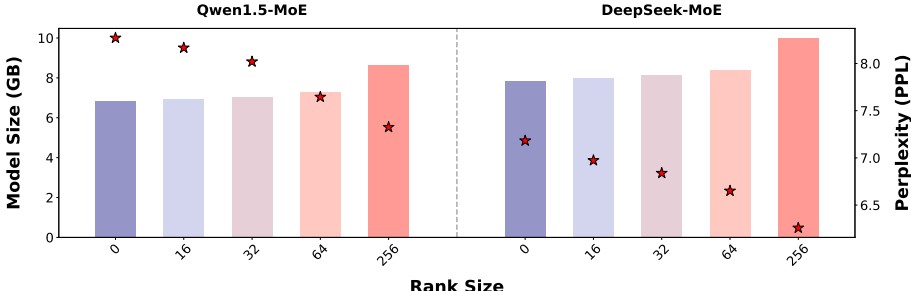

Figure 5: Comparison of Model Size (GB) and Perplexity (PPL) for DeepSeek-V2-Lite and Qwen1.5-MoE at 3-bit precision with different compensator ranks and a group size of 128. The $\star$ symbols indicate corresponding perplexity values on WikiText2.

### 3.2 ILP-BASED FRAMEWORK FOR UNIFIED GLOBAL ALLOCATION

Mixed-precision quantization and low-rank compensation are complementary techniques for compressing MoE models, yet integrating them within a single optimization framework remains nontrivial. To address the co-design challenges outlined in Section 3.1, we present BT-MoE, an ILP-driven framework that performs unified, global allocation of bit-widths and compensator ranks. BT-MoE formulates the joint selection as a constrained optimization problem over a discrete configuration space, enabling principled trade-offs between accuracy and resource usage and avoiding the suboptimality of heuristic decisions.

#### 3.2.1 OPTIMIZING UNIFIED (BIT, RANK) CONFIGURATIONS

The basic building block of our framework is the unified $(bit, rank)$ pair configuration assigned to each expert. To ensure each potential configuration is effective before our global search, we must optimally solve for its internal parameters. This involves finding the best possible balance between the precision offered by the bit-width and the error correction provided by the compensator rank.

To achieve this, we employ an iterative joint optimization process, inspired by MiLo (Huang et al., 2025a), which is superior to treating quantization and compensation as separate, sequential steps. For each candidate $(bit, rank)$ pair, we solve for the optimal quantization parameters(zero-point and scale) and compensator matrices $(U, V)$ by minimizing the overall representation error:

$$\arg\min_{z,s,U,V} \mathcal{L}(W - Q_{z,s}^{-1}(Q_{z,s}(W)) - UV).$$

This optimization alternates between two key steps:
**(1) Quantization Optimization:** For the given bit-width, we treat the compensated weight matrix $W - UV$ as the target for quantization. We then apply an iterative solver based on HQQ to this target matrix to find the optimal quantizer parameters $(z, s)$.
**(2) Low-Rank Compensator Optimization:** For the given rank, we use Singular Value Decomposition (SVD) on the current quantization residual $W - W_q$ to find the optimal compensator matrices

$(U, V)$. This process ensures that each configuration passed to our ILP solver is a locally-optimized implementation.

### 3.2.2 MODELING AS A MULTIPLE-CHOICE KNAPSACK PROBLEM

We model the task of assigning a $(bit, rank)$ configuration to each expert as a Multiple-Choice Knapsack Problem (MCKP). This analogy provides a clear and powerful framework for our optimization:

- The Knapsack represents the total allowed memory budget for the compression overhead.
- Each Expert in the MoE model corresponds to a group of items.
- Each possible $(bit, rank)$ Configuration for an expert is an item within that group (e.g., (3-bit, rank-32) is one item).
- The Value of an item is its contribution to accuracy, weighted by the **expert's importance**.
- The Weight of an item is its memory cost.

The objective is to select exactly one item (configuration) from each group (expert) to maximize the total value, without exceeding the knapsack's capacity (memory budget).

### 3.2.3 MEASURING CONFIGURATION IMPACT VIA LAYER-WISE QUANTIZATION LOSS

A prerequisite for our ILP formulation is an efficient method to quantify the impact of thousands of potential (bit, rank) configurations, especially for MoE models with a large number of experts. As a full model evaluation for each candidate is computationally prohibitive, we propose an efficient proxy metric based on layer-wise quantization loss. Our approach is based on a simple idea: when we compress a single expert, the error this creates mostly affects that expert's own layer. We measure this local error as the quantization error produced by compressing this expert.

Therefore, for each expert $e_i$ and every candidate configuration $c_j$, we conduct an isolated perturbation experiment. In this step, only the target expert is temporarily replaced with its compressed counterpart corresponding to the candidate configuration, while the remainder of the model is held constant in FP16. The quantization loss for this configuration, denoted $QuantLoss(e_i, c_j)$, is then formally defined as the Euclidean (L2) distance between the output of this perturbed layer and its corresponding FP16 reference. This systematic process is repeated for all experts and configurations, yielding a comprehensive sensitivity map that serves as the primary input for our ILP solver.

The quantization loss for this configuration, $L_{i,j}$, is then formally defined as the Euclidean (L2) distance between the output of this perturbed layer and its corresponding FP16 reference:

$$L_{ij} = ||\text{LayerOutput}(e_i \leftarrow c_j) - \text{LayerOutput}_{\text{FP16}}||_2.$$

This systematic process yields a comprehensive sensitivity map that serves as the primary input for our ILP solver, making the optimization computationally feasible.

### 3.2.4 INTEGER LINEAR PROGRAMMING FORMULATION

We formally define the optimization problem: Let $\mathcal{E} = \{e_1, e_2, \ldots, e_N\}$ be the set of $N$ experts in the model, and $\mathcal{C} = \{c_1, c_2, ..., c_K\}$ be the set of $K$ possible $(bit, rank)$ configurations. To find the optimal $(bit, rank)$ assignment, we formulate the problem as an optimization that seeks to minimize the total, importance-weighted quantization loss across all experts, subject to a strict memory budget. This allows us to systematically navigate the complex trade-offs between accuracy, expert sensitivity, and resource constraints.

**Parameters.**

- $F_i$: The activation frequency of expert $e_i$, serving as its importance weight.
- $L_{i,j}$: The layer-level quantization loss for expert $e_i$ under compression configuration $c_j$.
- $M_j$: The memory overhead incurred by adopting configuration $c_j$.
- $B$: The total memory budget for the compression overhead.

**Decision Variable.** We introduce a binary variable $x_{ij} \in \{0, 1\}$, where $x_{ij} = 1$ if expert $e_i$ is assigned configuration $c_j$, and 0 otherwise.

**Objective Function.** Our goal is to minimize the sum of importance-weighted and quantization losses across all experts:

$$\text{Minimize} \quad \sum_{i=1}^{N} \sum_{j=1}^{K} (F_i \cdot L_{ij}) \cdot x_{ij}. \tag{1}$$

**Constraints.** The optimization is subject to the following constraints:

- Unique Choice Constraint: Each expert must be assigned exactly one configuration

$$\sum_{j=1}^{K} x_{ij} = 1, \quad \forall i \in \{1, ..., N\}. \tag{2}$$

- Memory Budget Constraint: The total memory overhead from all chosen configurations cannot exceed the budget

$$\sum_{i=1}^{N} \sum_{j=1}^{K} M_j \cdot x_{ij} \leq B. \tag{3}$$

By adopting this two-stage approach, our framework fundamentally transforms the nature of the optimization problem. The initial brute-force search complexity of $O\left((|\mathcal{B}| \times |\mathcal{R}|)^{|\mathcal{E}|}\right)$, is entirely circumvented. Instead, our method's complexity is dominated by two stages:

**(1) Quant Loss Collection:** The first stage involves populating the sensitivity map. This requires conducting an isolated perturbation experiment for each configuration across all experts. The complexity of this stage is polynomial: $O\left(|\mathcal{B}| \times |\mathcal{R}| \times |\mathcal{E}|\right)$, which is practically feasible.

**(2) ILP Solving:** The second stage involves solving the formulated ILP problem. Our formulation maps the challenge onto a well-structured Multiple-Choice Knapsack Problem (MCKP). For such problems, modern, highly optimized solvers can employ sophisticated techniques such as branch-and-bound and cutting-plane methods to find the provably optimal solution in a time frame that is vastly more efficient than the original brute-force search. In practice, for all our experiments, the solver finds the optimal configuration in under 10 seconds.

Finally, we solve this ILP problem using Google OR-Tools (SCIP) (Perron & Didier), formulating the objective to minimize the total weighted quantization loss. This process yields the globally optimal (bit, rank) assignment for the entire MoE model.

## 4 EXPERIMENT

### 4.1 EXPERIMENT SETUP

**Models.** We evaluate BT-MoE on three open-source MoE models: Mixtral-8x7B (Jiang et al., 2024), DeepSeek-V2-Lite (Liu et al., 2024), and Qwen1.5-MoE (Team, 2024). DeepSeek-V2-Lite employs a hybrid architecture, using dense MLP instead of MoE blocks in the first layer. More details about the models are provided in the Appendix A.1.

**Baselines.** We evaluate BT-MoE against three representative quantization methods, all configured with a group size of 128 for fair comparison: **HQQ** (Badri & Shaji, 2023), a calibration-free method using half-quadratic quantization; **GPTQ** (Frantar et al., 2022), a calibration-based approach leveraging Hessian information for weight quantization; and **MiLo** (Huang et al., 2025a), a specialized MoE quantization method that utilizes low-rank compensators for extremely low-bit scenarios.

**Benchmarks.** We evaluate BT-MoE on five representative benchmarks, including Wikitext-2 (Merity et al., 2017), HellaSwag (Zellers et al., 2019), PIQA (Bisk et al., 2020), Lambada (Radford et al., 2019), and MMLU (Hendrycks et al., 2021). We present the performance on MMLU with 5-shot and others with zero-shot. All evaluations are conducted using the EleutherAI Language Model Evaluation Harness (Gao et al., 2024).

## 4.2 COMPARISON OF MODEL PERFORMANCE AND MEMORY FOOTPRINT

BT-MoE provides different compression configurations corresponding to various memory constraints. Each configuration individually sets the bit-width and compensator rank for every expert. Within these configurations, both the attention parts of the models and the shared experts in DeepSeek-V2-Lite and Qwen1.5-MoE are uniformly set to a (4-bit, rank=512) configuration.

Table 1: Model Perplexity and Accuracy Perplexity under Various Quantization Methods.

| Model | Method | Mem (GB) | WikiText2 (PPL) | HS | PQ | LO | MMLU | AVG |
|---|---|---|---|---|---|---|---|---|
| Mixtral-8x7B | FP16 | 88.90 | 3.700 | 86.02 | 83.67 | 80.87 | 71.34 | 80.48 |
| | GPTQ-3bit | 18.43 | 4.730 | 77.70 | 79.54 | 74.36 | 63.61 | 73.80 |
| | HQQ-3bit | 20.55 | 4.612 | 77.88 | 79.16 | 69.74 | 60.93 | 71.93 |
| | MiLo | 21.50 | 4.223 | 82.23 | 81.33 | 74.57 | 67.07 | 76.30 |
| | BT-MoE | 18.37 | 4.480 | 82.01 | 81.23 | 73.70 | 63.44 | 75.10 |
| | BT-MoE | 20.36 | 4.095 | 84.64 | 83.03 | 76.88 | 66.76 | 77.83 |
| DeepSeek-MoE | FP16 | 31.24 | 5.832 | 77.33 | 79.00 | 73.88 | 45.07 | 68.82 |
| | GPTQ-3bit | 6.97 | 6.843 | 70.98 | 76.44 | 68.62 | 32.53 | 62.14 |
| | HQQ-3bit | 7.67 | 7.082 | 71.38 | 77.25 | 66.67 | 35.63 | 62.73 |
| | MiLo | 8.18 | 6.423 | 74.15 | 78.12 | 71.47 | 41.97 | 66.42 |
| | BT-MoE | 6.76 | 6.640 | 72.32 | 77.86 | 70.95 | 41.41 | 65.64 |
| | BT-MoE | 8.18 | 6.180 | 75.51 | 78.83 | 73.51 | 42.93 | 67.70 |
| Qwen1.5-MoE | FP16 | 26.70 | 6.521 | 77.86 | 81.25 | 71.90 | 62.50 | 73.38 |
| | GPTQ-3bit | 6.73 | 8.293 | 72.77 | 76.80 | 62.33 | 54.36 | 66.56 |
| | HQQ-3bit | 6.83 | 8.272 | 72.46 | 77.15 | 62.48 | 51.29 | 65.85 |
| | BT-MoE | 6.83 | 7.377 | 74.54 | 78.31 | 67.75 | 56.71 | 69.35 |

To ensure fairness, we compare the accuracy of quantized models with different methods under similar memory footprints. The comprehensive results, summarized in Table 1, show that BT-MoE demonstrates superior performance compared to all baselines across all evaluated models.

On Mixtral-8x7B, BT-MoE shows significant advantages. At a memory footprint of 18.37 GB, our method surpasses the slightly larger GPTQ-3bit baseline (18.43 GB) by over 1.3 points in average accuracy. More impressively, our 20.36 GB configuration drastically outperforms HQQ-3bit (20.55 GB), reducing the WikiText2 perplexity from 4.612 to 4.095 and increasing the average score from 71.93 to 77.83, recovering a substantial portion of the performance gap to the FP16 model. This result exemplifies our framework's ability to establish a better accuracy-memory trade-off, with consistent advantages observed across all tested models. More detailed and complete experimental results are presented in the Appendix A.1.4.

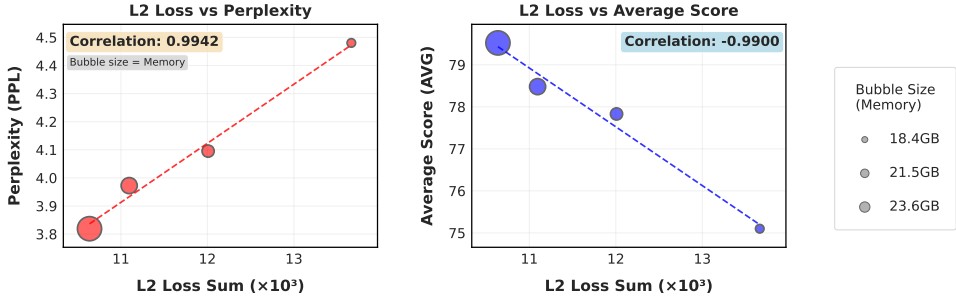

Figure 6: The Objective Function Value as a proxy metric for final model performance on Mixtral-8x7B. The bubble size represents the memory footprint of the compressed model under each quantization configuration.

We validate that our ILP objective function, the Importance-Weighted L2 Loss Sum, serves as a highly reliable proxy for final model performance. As shown in Figure 6, this predicted loss exhibits a remarkable correlation of 0.9942 with Perplexity and -0.9900 with Average Score on various datasets, confirming its effectiveness in guiding our optimization.

### 4.3 ABLATION STUDY

To quantify the individual contributions of mixed-precision and low-rank compensation within our framework, we conduct an ablation study comparing our full BT-MoE method against two ablated variants that represent each strategy in isolation. In Figure 7, the **Base** refers to the uniform-bit quantization method. The **Base+Mix-Precision** represents a mixed-precision-only approach, which allocates bit-widths based on an activation frequency heuristic without any compensators. The **Base+Compensation** represents a compensation-only approach, applying low-rank compensators to a uniform bit-width.

The results in Figure 7 reveal that the quality of heuristic-based mixed-precision is critically dependent on whether the heuristic itself can accurately measure expert importance. A clear example of this limitation is on the Mixtral-8x7B model, where the frequency-based method fails to outperform the uniform-4bit HQQ baseline. As we discussed in Section 3.1, this is mainly because Mixtral8x7B features a small number of experts with low variance in their activation frequencies, making frequency an unreliable proxy for expert importance.

In contrast, our joint optimization method BT-MoE consistently achieves the best accuracy-memory trade-off. On the DeepSeek-MoE model, for instance, while **Base+Compensation** achieves strong accuracy, BT-MoE delivers comparable or better performance with a significantly smaller memory footprint. This demonstrates that by jointly optimizing both bit-width and rank, BT-MoE can navigate the complex trade-offs that simple heuristics cannot, validating the superiority of our global optimization approach.

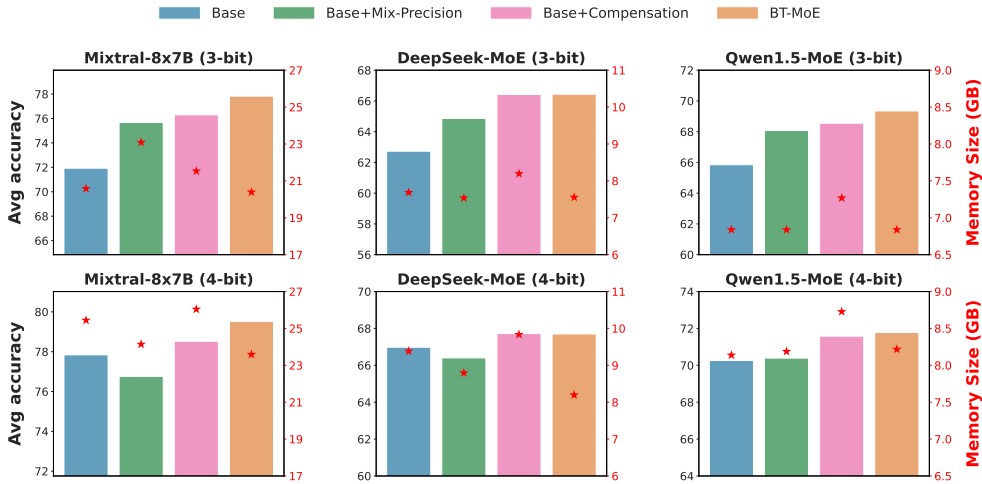

Figure 7: Results of different quantization methods and ours BT-MoE on various MoE models. The ⋆ symbols denote the memory size of the compressed model.

## 5 CONCLUSION

In this study, we address the critical challenges of deploying MoE models with extreme compression. We introduce BT-MoE, a novel framework that unifies the allocation of mixed-precision bit-widths and low-rank compensators into a tractable Integer Linear Programming problem. We design an objective function to minimize the importance-weighted sum of quantization losses, which we validate as a highly reliable predictor of final model performance. This budget-aware tuning approach systematically discovers a globally optimal configuration that maximizes model accuracy under a strict memory budget. Our extensive experiments demonstrate that it consistently outperforms existing quantization techniques by achieving higher accuracy at similar or even lower memory footprints. These results highlight the potential of our unified optimization strategy to make the deployment of large MoE LLMs more feasible in resource-constrained environments.

STATEMENT

ETHICS STATEMENT

This research was conducted using publicly available, open-source models and standard academic benchmarks, and did not involve the use of human subjects or private user data. Our work introduces BT-MoE, a compression technique for MoE models. Any ethical risks, such as potential biases or the generation of harmful content, are therefore inherited from the original models, as our work does not create new models.

REPRODUCIBILITY STATEMENT

To ensure the reproducibility of our work, we provide detailed information regarding the code, models, datasets, and experimental setup used in this paper. We also provide a README file with detailed reproduction instructions in our code submission.

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

# A APPENDIX

## A.1 FULL EXPERIMENTAL RESULTS

### A.1.1 MODEL DETAILS IN EXPERIMENT

Table 2: Architectural Specifications of Evaluated MoE Models

| Model | Params (GB) | Experts | TopK |
|---|---|---|---|
| Mixtral-8×7B | 92.9 | 8 | 2 |
| Qwen1.5-MoE | 26.7 | 60 + 4 | 4 |
| DeepSeek-V2-Lite | 29.3 | 64 + 2 | 6 |

### A.1.2 THE EFFECT OF COMPENSATOR RANK ON THE ACCURACY RECOVERY OF LOW-BIT QUANTIZED WEIGHTS

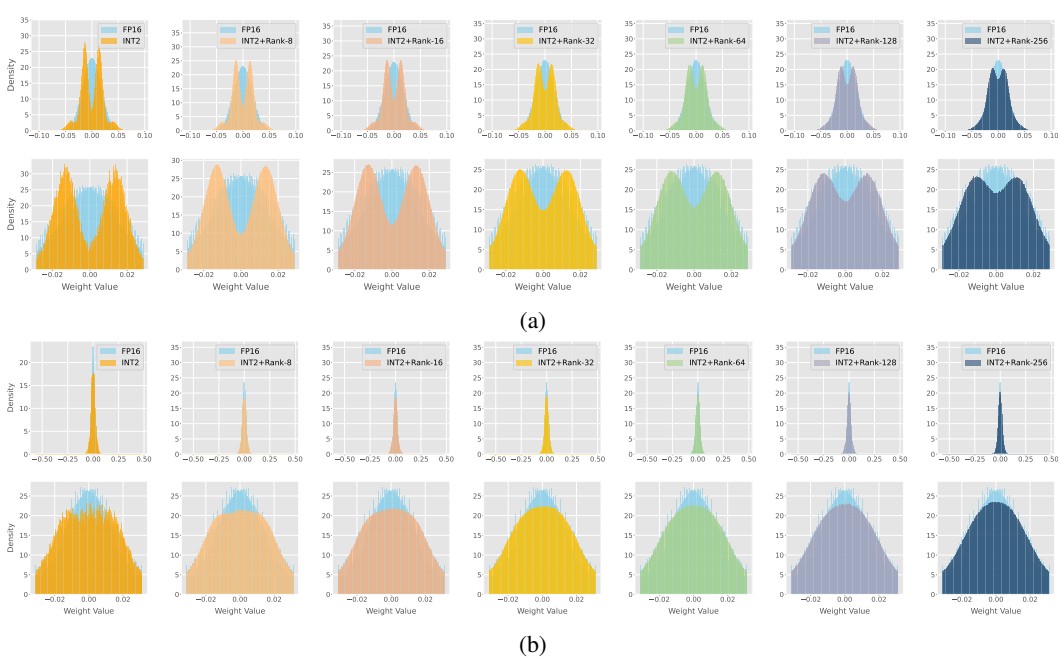

Figure 8: The progressive recovery of weight distributions for different model components under aggressive INT2 quantization. The top set of plots (a) shows the distributions for an expert's weights, while the bottom set (b) shows distributions for attention weights. Each column corresponds to a different compensator rank, from Rank-0 to Rank-256.

### A.1.3 OFFLINE COMPUTATIONAL COST FOR DATA COLLECTION

Our framework requires a one-time, offline data collection phase to generate the sensitivity map (the $L_{i,j}$ values) used by the ILP solver. This involves performing an isolated perturbation experiment for every expert across all candidate $(bit, rank)$ configurations, as described in Section 3.2.3.

On a single NVIDIA A100 GPU, this data collection process takes approximately 4 to 12 hours, with the duration depending on the total number of experts in the MoE model (e.g., Mixtral-8x7B is on the lower end, while DeepSeek-MoE is on the higher end). It is important to note that while this upfront data collection is computationally intensive, it is a one-time cost. Once this comprehensive sensitivity map is generated, the ILP optimization itself is extremely efficient, consistently finding the globally optimal configuration in under 10 seconds.

### A.1.4 FULL EXPERIMENTAL RESULTS UNDER DIFFERENT MEMORY CONSTRAINTS

Table 3: The full results of GPTQ, HQQ, MiLo, MPQ-Freq and ours BT-MoE with 3-bit and 4-bit weight quantization among 5 datasets on Mixtral-8x7B, DeepSeek-MoE-16B and Qwen1.5-MoE-14B.We evaluate on the following datasets: Wikitxt-2, HellaSwag(HS), LAMBADA-openai(LO), PIQA(PQ) and MMLU. Specifically, the GPTQ method utilizes Wikitext2 as the calibration dataset. MPQ-Freq is a heuristic mixed-precision-only strategy, which allocates bit-widths based on activation frequency without any compensators.

| Model | Bits | Memory | WikiText2(PPL) | HS | PQ | LO | MMLU | AVG |
|---|---|---|---|---|---|---|---|---|
| Mixtral-8x7B | FP16 | 88.90G | 3.700 | 86.02 | 83.67 | 80.87 | 71.34 | 80.48 |
| | GPTQ-4bit | 23.81G | 4.233 | 80.73 | 81.44 | 74.42 | 67.84 | 76.11 |
| | GPTQ-3bit | 18.43G | 4.731 | 77.70 | 79.54 | 74.36 | 63.61 | 73.80 |
| | HQQ-4bit | 25.41G | 3.941 | 83.65 | 82.62 | 76.57 | 68.53 | 77.84 |
| | HQQ-3bit | 20.55G | 4.612 | 77.88 | 79.16 | 69.74 | 60.93 | 71.93 |
| | MiLo | 21.50G | 4.223 | 82.23 | 81.33 | 74.57 | 67.07 | 76.30 |
| | MiLo | 26.01G | 3.774 | 85.47 | 83.18 | 77.31 | 68.13 | 78.52 |
| | MPQ-Freq | 23.06G | 4.777 | 82.02 | 82.37 | 72.23 | 66.07 | 75.67 |
| | MPQ-Freq | 24.11G | 4.354 | 82.77 | 82.53 | 74.70 | 67.07 | 76.76 |
| | BT-MoE | 18.37G | 4.4803 | 82.01 | 81.23 | 73.70 | 63.44 | 75.10 |
| | BT-MoE | 20.36G | 4.0953 | 84.64 | 83.03 | 76.88 | 66.76 | 77.83 |
| | BT-MoE | 21.50G | 3.9725 | 85.04 | 83.08 | 77.99 | 67.79 | 78.48 |
| | BT-MoE | 23.56G | 3.8191 | 85.57 | 83.51 | 79.39 | 69.60 | 79.52 |
| DeepSeek-MoE | FP16 | 31.24G | 5.832 | 77.33 | 79.00 | 73.88 | 45.07 | 68.82 |
| | GPTQ-4bit | 8.75G | 6.266 | 74.67 | 78.14 | 72.28 | 42.23 | 66.83 |
| | GPTQ-3bit | 6.97G | 6.843 | 70.98 | 76.44 | 68.62 | 32.53 | 62.14 |
| | HQQ-4bit | 9.37G | 6.187 | 74.68 | 78.61 | 72.32 | 42.30 | 66.98 |
| | HQQ-3bit | 7.67G | 7.082 | 71.38 | 77.25 | 66.67 | 35.63 | 62.73 |
| | MiLo | 8.18G | 6.423 | 74.15 | 78.12 | 71.47 | 41.97 | 66.42 |
| | MiLo | 9.82G | 5.946 | 76.74 | 78.83 | 72.11 | 43.21 | 67.72 |
| | MPQ-Freq | 7.52G | 6.543 | 72.45 | 78.00 | 68.95 | 40.05 | 64.86 |
| | MPQ-Freq | 8.78G | 6.319 | 74.39 | 78.07 | 70.95 | 42.19 | 66.40 |
| | BT-MoE | 6.76G | 6.640 | 72.32 | 77.86 | 70.95 | 41.41 | 65.64 |
| | BT-MoE | 7.54G | 6.348 | 73.71 | 78.13 | 71.94 | 41.98 | 66.44 |
| | BT-MoE | 8.18G | 6.180 | 75.51 | 78.83 | 73.51 | 42.93 | 67.70 |
| Qwen1.5-MoE | FP16 | 26.70G | 6.521 | 77.86 | 81.25 | 71.90 | 62.50 | 73.38 |
| | GPTQ-4bit | 8.03G | 7.544 | 75.61 | 78.40 | 64.46 | 58.25 | 69.18 |
| | GPTQ-3bit | 6.73G | 8.293 | 72.77 | 76.80 | 62.33 | 54.36 | 66.56 |
| | HQQ-4bit | 8.13G | 7.143 | 75.60 | 78.65 | 67.22 | 59.58 | 70.26 |
| | HQQ-3bit | 6.83G | 8.272 | 72.46 | 77.15 | 62.48 | 51.29 | 65.85 |
| | MiLo | 7.26G | 7.326 | 75.24 | 78.40 | 66.28 | 55.28 | 68.55 |
| | MiLo | 8.72G | 6.860 | 77.32 | 79.64 | 70.99 | 58.45 | 71.58 |
| | MPQ-Freq | 6.83G | 7.667 | 73.81 | 78.00 | 65.44 | 55.08 | 68.08 |
| | MPQ-Freq | 8.18G | 7.062 | 75.32 | 78.40 | 69.60 | 58.22 | 70.39 |
| | BT-MoE | 6.83G | 7.377 | 74.54 | 78.31 | 67.75 | 56.71 | 69.35 |
| | BT-MoE | 8.21G | 6.880 | 76.37 | 79.43 | 71.16 | 60.17 | 71.78 |

### A.2 THE USE OF LARGE LANGUAGE MODELS (LLMs)

During the preparation of this manuscript, the authors utilized Large Language Models (LLMs) as a writing assistant. The primary use of these tools was for language enhancement, including improving grammar, clarity, and readability through translation and polishing of the text.

