# OpenReview forum: "BT-MoE: A Budget-Aware Tuning Framework for Joint Bit–Rank Allocation in MoE Models"
_ICLR.cc/2026/Conference — ICLR 2026 Conference Withdrawn Submission_

### Official Review · Reviewer_ApXF · 2025-10-19

**Soundness:** 3
**Presentation:** 3
**Contribution:** 2
**Rating:** 4
**Confidence:** 4

**Summary:**

This paper proposes the joint optimization of bit width and compensator rank. The key observations are that both the bit width for MoE model quantization and the compensator rank affect the model performance, and they exhibit significant differences across different experts. Additionally, it is observed that there is a dependency between the two, so they cannot be optimized independently. To address this joint optimization problem, the paper proposes the following approaches: Using a proxy loss instead of directly measuring the model loss, where the proxy loss is sampled independently for each layer and each expert. Formulating the optimization problem as an ILP problem and then solving it under a given memory constraint. The paper uses the above techniques to test perplexity on WikiText2 and also evaluates performance on benchmarks such as HS, PQ, LO, and MMLU. The results show that its performance outperforms that of GPTQ-3bit, HQQ-3bit, and MiLo across all these benchmarks.

**Strengths:**

1.	The paper identifies the complex and distinct requirements for bitwidth and compensator rank among different experts in MoE models during quantization.
2.	The paper effectively formulates the problem as an ILP problem via the proxy loss and solves it.
3.	The paper experimental performance is better than that of GPTQ-3bit, HQQ-3bit, and MiLo.

**Weaknesses:**

1.	The design space of proxy loss is not well explained and other potential options are not evaluated.
2.	The formulations in this paper need more accurate explanations.
3.	More details about the optimization process are needed.
4.	The evaluation is mainly for small MoE models and the performance for large MoE models is not revealed.
5.	Comparison to important related-work is missing.

**Questions:**

1.	Proxy Metric Robustness: The paper validates the proxy metric via correlation with perplexity/average score, but it does not compare this proxy to alternative metrics (e.g., KL divergence). Why is L2 distance the best choice for capturing layer-level error? A short ablation comparing proxy metrics would strengthen this design choice.
2.	There lacks explanation about how to calculate Mj for each cj, which is used in equation (3).
3.	The joint optimization of (z,s) and (U,V) are done in an iterative manner, but the hyper-parameters such as the optimization steps and convergence criteria are missing.
4.	How is the performance of BT-MoE for large MoE models?
5.	The paper mentioned MxMoE as a mixed precision MoE method, but didn’t compare to it in evaluation.

---

### Official Review · Reviewer_sGav · 2025-10-28

**Soundness:** 2
**Presentation:** 4
**Contribution:** 2
**Rating:** 4
**Confidence:** 5

**Summary:**

Proposed BT-MoE jointly optimizes bit-width allocation and low-rank compensator for MoE expert compression by framing it as a Multiple-Choice Knapsack Problem. The proposed method employ layer-wise quantization loss as a proxy and an ILP solver to find the optimal trade-off between model accuracy and memory footprint.

**Strengths:**

This paper is easy to understand, and the joint-optimization problem is well defined. The solution expands the search space compared to previous work and it seems to work (with a better benchmark performance) according to the experiment results.

**Weaknesses:**

The proposed method appears to be a straightforward combination of existing techniques: mixed-precision allocation from MxMoE and low-rank compensation from MiLo. The "joint optimization" merely expands the search space for the ILP solver without introducing any substantive technical challenges or novel contributions to either the bit-width allocation or low-rank compensation problems individually.

**Lack of Efficiency Evaluation**: For a work focused on efficiency, the complete absence of practical performance experiments is a critical flaw. The paper provides no data on kernel speed or end-to-end throughput, making it impossible to verify if the theoretical memory savings translate into actual runtime improvements.

**Questions:**

1. Could you please provide efficiency estimation or evaluation？
2. Inter-layer dependency is well studied in model quantization field but is simply ignored in this work (Line293-295). Have you tested how much this affects the performance?

---

### Official Review · Reviewer_cvMu · 2025-10-30

**Soundness:** 3
**Presentation:** 3
**Contribution:** 2
**Rating:** 4
**Confidence:** 3

**Summary:**

This paper presents the BT-MoE framework, which offers a novel approach to efficiently deploying MoE models by addressing the challenges of quantization and low-rank compensation. The method combines mixed-precision quantization and low-rank compensators in a unified optimization framework. Specifically, it frames the joint design of mixed-precision bit-widths and compensator ranks as a MCKP, enabling an optimal configuration under a strict memory budget. The authors employ a proxy metric based on layer-wise quantization loss, which makes the optimization computationally feasible.

**Strengths:**

1. The paper is well-structured and clearly written, making complex ideas like mixed-precision quantization and low-rank compensators easy to understand through concise explanations and intuitive visual aids.

2. The motivation of the paper is clearly articulated and convincing, effectively highlighting the challenges and the need for the proposed solution in optimizing MoE models under memory constraints.

**Weaknesses:**

1. Novelty Concern: The main contribution of the paper seems to be modeling the search for mixed-precision bit-width allocation and low-rank compensation for MoE models as an ILP problem. However, many of the other aspects of the approach appear to rely on existing methods and prior work. Could you elaborate on the novelty of the overall approach and how it advances beyond the current state of the art in MoE model compression?
2. Regarding Expert Activation Frequency Calibration: In the paper, expert activation frequency ($F_i$ in Eq. (1)) requires calibration using data. Could you clarify the sample size used for this calibration and the sequence length (seqlen)? Additionally, since expert activation responses may vary based on the dataset or task, how robust are the results when applied to different datasets?
3. Choice of L2 Loss as Proxy Loss: Why was L2 loss chosen as the proxy loss in the paper, rather than other metrics such as MSE or KL divergence? I’m curious to understand the reasoning behind this choice.
4. Mixed-Precision Quantization in Figure 7: In Figure 7, the mixed-precision quantization with 4-bit for the Mixtral and DeepSeek models results in lower accuracy than the uniform bit-width quantization. This seems somewhat counterintuitive, as mixed-precision quantization, with reasonable bit allocation, is typically expected to perform better than uniform bit-width quantization with the same average bit-width. Could you provide some insights into this discrepancy?
5. Layer Comparison in Experts: Since different layers have varying impacts on the final result, is it reasonable to compare experts from different layers together using a unified approach? I’m curious if this might affect the interpretation of the results.
6. Absence of Bit-Width Comparison and Acceleration Effects: The tables in the experiments appear to lack comparisons between different bit-widths for the methods, as well as a discussion on acceleration effects. Could you provide some clarification on why these comparisons were not included?

**Questions:**

See Weaknesses

---

### Official Review · Reviewer_cG1X · 2025-11-01

**Soundness:** 2
**Presentation:** 2
**Contribution:** 2
**Rating:** 2
**Confidence:** 4

**Summary:**

This paper presents BT-MoE, a novel framework for compressing MoE models. The core idea is to jointly optimize the mixed-precision bit-width and low-rank compensator rank for each expert under a global memory budget. The authors effectively formalize this challenging co-design problem as a MCKP. To make this NP-hard problem tractable, they introduce an efficient layer-wise quantization loss as a proxy metric and use an ILP solver to find the global optimal configuration. Comprehensive experiments on several MoE models (Mixtral, DeepSeek, Qwen) demonstrate that BT-MoE consistently outperforms strong baselines, achieving a superior accuracy-memory trade-off, especially in aggressive sub-4-bit scenarios.

**Strengths:**

1. Novel and Well-Motivated Formulation: The key insight of formalizing the joint bit-width and rank allocation problem as a MCKP and modeling it as an Integer Linear Programming problem is both novel and well-motivated. This formulation elegantly captures the complex, coupled relationship between these two compression dimensions, representing a significant advancement over prior works that treat them independently.

2. Practical and Efficient Quantization Loss Computation: The use of a layer-wise quantization loss as a proxy metric avoids the prohibitive cost of full-model evaluation for each candidate configuration. Its combination with a standard ILP solver makes the global optimization computationally feasible and reproducible.

3. Strong Empirical Validation: This paper provides extensive experiments across multiple state-of-the-art MoE models and benchmarks. The results clearly demonstrate that BT-MoE achieves superior performance compared to established baselines such as GPTQ, HQQ, and MILo, particularly at low bit-widths. The conducted ablation studies effectively validate the importance of the joint optimization.

**Weaknesses:**

1. Limited Novelty: The paper primarily establishes an ILP formulation to solve the joint bit-width and rank allocation problem, and proposes a layer-wise quantization loss to make the optimization tractable. However, the optimization error potentially introduced by this proxy metric is not thoroughly discussed, nor do the authors explore whether more computationally efficient or effective quantization losses exist. Furthermore, the quantization process directly reuses MiLo's pipeline without novel discussion or improvement.
2. Justification of Layer-wise Quantization Loss: The primary foundation for making the MCKP problem tractable is the substitution of the global quantization loss with a local, layer-wise loss. However, the paper lacks an in-depth discussion on the theoretical justification for this substitution. While a correlation analysis between this loss and PPL is provided, analysis with other key performance metrics is absent.
3. Experimental results could be more comprehensive: The current baselines only include MiLo as a method utilizing low-rank compensators, which are known to significantly bridge the accuracy gap. The experimental section could be strengthened by introducing more powerful baselines that also incorporate low-rank compensators. Furthermore, employing the same quantization strategy as MiLo potentially undermines the persuasiveness of the results demonstrating superiority over it.
4. The experimental reliability is questionable. In Table 1, the DeepSeek-MoE results align exactly with the MMLU scores reported in MiLo’s Table 3, yet the other results have significantly worsened.

**Questions:**

1. Justification of the Layer-wise Quantization Loss: Could the authors provide further justification, either from a theoretical or experimental perspective, for the use of this layer-wise quantization loss to simplify the problem and make it tractable?
2. More Comprehensive Experiments: Could the authors provide more comprehensive experiments, such as including additional baselines that utilize low-rank compensators or even non-heuristic quantization methods, to more thoroughly demonstrate the effectiveness of the proposed approach?
3. Quantization Bias Towards PPL: The paper highlights a significant correlation between the layer-wise quantization loss and PPL. Could the authors clarify whether this strong correlation might lead the quantization process to exhibit an excessive or disproportionate focus on optimizing for PPL, potentially at the expense of other important performance metrics?

4. Offline Computational Overhead：The offline data collection phase is a bottleneck. Have the authors explored or considered methods to reduce this cost, for instance, through sampling a subset of experts/layers or using a more efficient proxy?

5. Could the authors provide a serious explanation as to why, if some entries align perfectly with MiLo’s Table 3, the remaining baseline results deteriorate so noticeably?

---

### Note · Authors · 2025-12-23

I have read and agree with the venue's withdrawal policy on behalf of myself and my co-authors.